# Badder Seeds: Reproducing the Evaluation of Lexical Methods for Bias Measurement

## Reproducibility Summary

**Scope of Reproducibility**

Combating bias in NLP requires bias measurement. Bias measurement is almost always achieved by using *lexicons of seed terms*, i.e. sets of words specifying stereotypes or dimensions of interest. This reproducibility study focuses on Antoniak and Mimno [1]'s main claim that the rationale for the construction of these lexicons needs thorough checking before usage, as the seeds used for bias measurement can themselves exhibit biases. The study aims to evaluate the reproducibility of the quantitative and qualitative results presented in the paper and the conclusions drawn thereof.

**Methodology**

We re-implement the entirety of the approaches outlined in the original paper. We train a skip-gram word2vec model with negative sampling to obtain embeddings for four corpora. This does not require particular computing requirements beyond standard consumer personal computers. Additional code details can be found in our linked repository.

**Results**

We reproduce most of the results supporting the original authors' general claim: seed sets often suffer from biases that affect their performance as a baseline for bias metrics. Generally, our results mirror the original paper's. They are slightly different on select occasions, but not in ways that undermine the paper's general intent to show the fragility of seed sets.

**What was difficult**

The significant difficulties encountered were due to a lack of publicly available code and documentation to clarify missing information in the paper. For this reason, many algorithms that ultimately turned out to be quite simple required lengthy clarifications with authors or trial and error. Lastly, the research was quite data-intensive, which caused some implementations to be non-trivial to account for memory management.

**What was easy**

Once understood, the methods proposed by the authors were relatively easy to implement. The mathematics involved is quite straightforward. Communication was also reasonably accessible. The authors' emails were readily available, and the responses came quickly and were always helpful.

**Communication with original authors**

We maintained a lengthy email correspondence throughout the replication of the paper with one author, Maria Antoniak. We contacted her to clarify extensive aspects of the paper's methodology. Specifically, this concerned summarizing the data processing approach, explaining missing hyperparameters, and outlining the aggregation of metrics across different bootstrapped models. None of the original code was disclosed.

# 1 Introduction

The emergence of bias quantification in Natural Language Processing (NLP) methods has given rise to two use cases, referred to as *downstream* and *upstream*. In the former, bias measurements are used to debias or correct biases in word representations to avoid encoded biases trickling down when applying these NLP models [2, 3]. In the latter, bias measurements are used on models trained on small corpora to quantify the bias present and compare them. This use case has endowed social scientists with the quantitative foundation to answer political and social questions about bias across corpora in an empirical manner. [11, 6] Crucially, most bias quantification methods depend on lexicons of seed terms that specify the bias dimensions of interest. The selection of seed terms varies considerably across the literature, and seed sets themselves may exhibit social and cognitive biases [1]. It is not clear whether it is possible to re-use seed set across corpora (thereby interfering with *upstream* use cases), and elements such as seed term frequency have been shown to affect bias measurements, and thus *downstream* uses [4].

We seek to replicate the Antoniak and Mimno [1] paper, hereafter referred to as "the original paper/work". In it, the authors seek to 1) qualitatively explore seed selection and their sources, 2) demonstrate that features of seed sets such as pairing order, set similarity, and frequency can cause instability in bias measurements, and 3) make recommendations for the testing and justifying of seed sets in future work. We have replicated the experiments showing the fragility of seed sets, thus verifying the claims of a need for better justification and analysis of them in future literature. We have also built a public toolkit to reproduce these measures on arbitrary seed sets and trained embeddings.

# 2 Scope of reproducibility

This reproducibility study focuses on the authors' main claim that seed lexicons need thorough checking before usage to measure bias, as seeds themselves can be biased and induce instabilities in measurement. The authors conducted a literature review on prior works to gather many seed sets. They subsequently evaluated the gathered seed sets with a series of bias measurement metrics proposed by Bolukbasi et al. [2], Caliskan et al. [3], and themselves.

Our work consists of two interconnected efforts: code replication, given the absence of pre-existing code for the original paper, and reproducing the main results. The latter goal is the main focus of our work and entails reproducing the outcomes that support the paper's central claims, which can be summarized as follows:

1. Bias subspaces generated from common bias subspace metrics (e.g., WEAT, PCA) can help capture the difference represented by the seed set pairs.

2. Bias subspaces suffer from instability due to the following factors:

    (a) The ordering and pairing of the seed sets.
    (b) The selection of seeds that are members of the seed sets.
    (c) The degree of semantic similarity between seeds.

3. Methods of sourcing seed sets are inconsistent, with disparate strategies being used across NLP literature.

# 3 Methodology

The code from the original paper was not made publicly available. We, therefore, re-implemented the entire approach from the description in the original paper. The following section will summarize the resources and methodology used to reproduce the original paper accurately.

## 3.1 Code

As mentioned above, the code from the original paper is not publicly available. We fully re-implement all the code, which can be found on GitHub[1]. We closely follow the original paper's methodology to achieve accurate reproduction. The reproduction is performed step by step, from downloading and preprocessing the data to training the models and visualizing the results.

---

[1]https://anonymous.4open.science/r/mlrc-2021-A0C2

### 3.2 Documentation

Unfortunately, there was little to no documentation in the original work besides the content of the original paper. This occasionally lacked crucial information to reproduce the results or was vague on implementation details. In addition to the original paper, Antoniak and Mimno [1] published a Github repository that contained a JSON with the metadata on seed sets gathered from prior works [2].

### 3.3 Model descriptions

We train several bootstrapped skip-gram word2vec models with negative sampling on unigrams on each dataset. This model attempts to predict whether a particular word is a valid context (where the context window size is a hyperparameter) for a given other word using a single fully connected hidden layer. The first step in training this model is creating a vocabulary of the entire training dataset. With this vocabulary, each word can be represented as a one-hot vector. The network output is then a measure of the probability that the word is a valid context. The trained weights from this hidden layer are then used to obtain word embedding vectors for each term in the training set vocabulary.

### 3.4 Datasets

The original paper used four datasets and one pretrained model: New York Times articles from April 15th-June 30th, 2016[3]; high-quality WikiText articles, using the complete WikiText-103 training set [7]; Goodreads book reviews for the romance and history and biography genres sampled from the UCSD book Graph [12, 13]; and the pretrained word2vec GoogleNews model [4]. We use these same corpora for our research, preprocessing them as closely as possible to the original paper. This consists of grouping the text into documents, filtering relevant documents, lowercasing and removing special characters. We then use spaCy [5] for tokenization and POS-tagging. Because the work is not concerned with model performance, this study makes no use of train/dev/test splits. The WikiText-103 dataset, however, is pre-split, so like in the original work, we work with the training split. Links to all these datasets can be found in our Github repository.

Preprocessing statistics of our work and the original paper can be found in Table A.1. We find general agreement in our numbers regarding the total number of documents per dataset. There are minor discrepancies in the Goodreads datasets, most likely due to implementation differences. We also count slightly fewer total words than the original paper in all cases, but the orders of magnitude generally match. We are, however, unable to reproduce vocabulary size accurately. We tried many strategies in the replication process to obtain these numbers, but none were successful. Furthermore, looking at the official dataset statistics, for example for WikiText [7], it is clear that our reproduced vocabulary size is a lot closer to the ground truth than the one by Antoniak and Mimno [1]. Lastly, mean document length values of each dataset are accurately reproduced, with the WikiText values suffering the most. The subsections below will discuss each dataset in more detail.

**New York Times**   This dataset contains 165,900 paragraphs from 8,888 articles from the New York Times published between April 15th and June 30th 2016. The articles cover a broad range of sections, including but not limited to movies, sports, technology, business, books, science, and fashion.

**WikiText-103**   This dataset contains 28,472 manually verified articles from Wikipedia.org. The entire training dataset is used, in which lists, HTML errors, math, and code have already been removed. Furthermore, we removed all formulas still present in the text.

**Goodreads**   The entire Goodreads dataset contains millions of reviews. This study uses just the Romance and the History/Biography genres. Five hundred book reviews per book are sampled for each genre while filtering out all books with fewer than 500 reviews and all reviews containing fewer than 20 characters.

**GoogleNews**   Google's pretrained word2vec model is trained on ca.100 billion words from the GoogleNews dataset (4). Our use of this model was limited to replicating the results outlined below for additional robustness.

---

[2] https://github.com/maria-antoniak/bad-seeds
[3] https://www.kaggle.com/nzalake52/new-york-times-articles
[4] https://github.com/mmihaltz/word2vec-GoogleNews-vectors

**Seed Set Dataset** Part of the contributions of the original work was creating a catalogue of 178 seed sets gathered from eighteen highly-cited prior works on bias measurements. We refer to this catalogue as the *gathered seeds*. Each element of the catalogue comprises a seed set, the category it represents, a justification, the source categorization, a link, and a unique ID. It is readily available on the original author's GitHub[2]. A brief statistical overview can be found in Fig. A.1. We process the catalogue by lower-casing the seeds and removing bigrams to use them with our models. We also filter seed sets containing less than two seeds as we argue that a single seed would not be sufficient to form a set.

## 3.5 Experimental setup and code

An environment containing all necessary packages is included in the publicly available repository and can be quickly set up. To mirror the original paper's setup, we used the *gensim* [10] implementation of skip-gram with negative sampling [8] to train the vector embeddings for all datasets. We used this library to train our models as that is the framework used by the original paper and to avoid noise due to different implementations (the investigation of which would be outside the scope of this paper). Several PyTorch [9] implementations are also available on GitHub if that is preferred. [5,6]

We reproduce the original paper's results by focusing on two popular seed-based bias metrics to measure bias in corpus-derived embeddings: WEAT and PCA. These metrics are used to produce a *bias subspace* vector given a pair of seed sets that specifies a bias dimension of interest. The WEAT method, introduced in Caliskan et al. [3], produces a vector based on the difference between the mean vectors of the two target sets. The PCA method, described in Bolukbasi et al. [2], instead requires that each seed term in one of the seed sets be paired with one seed term from the other seed set. The subspace vector is then the first principal component resulting from the PCA of a matrix constructed by, for each pair of seeds, taking the two half vectors from the pair's mean to the two pair members and using them as two columns of the matrix.

We also reproduce the original paper's coherence metric, which aims to quantify the robustness of the bias subspace. This metric is calculated as the absolute value of the difference in mean ranks of the terms in two seed sets when all the model's vocabulary is ranked by cosine similarity to the bias subspace. Another metric used is set similarity, the cosine similarity between the average vectors of two seed sets.

Finally, when aggregating embeddings of a specific word across bootstrapped models, we take the average of the embedding vectors in each model that includes the word. Given a particular pair of seed sets for coherence aggregation, we only average coherence scores for models containing every seed term in the two sets to avoid aggregating coherence based on different seed sets.

## 3.6 Hyperparameters

100-dimensional embeddings were trained for five epochs on all four datasets, with a five-word negative context sampling rate and a window size of five. We trained embeddings with a minimum word count of 0, 10, and 100 due to variation in the original paper. This process was repeated for 20 bootstrapped samples of each dataset (with the sample size equal to the number of documents in the dataset), resulting in 20 separate models. The bootstrapping provided the stochasticity required for robustness. To ensure this reproducibility, we use a random seed of 42 throughout.

## 3.7 Computational requirements

The execution of the reproduced code does not take excessive computing power. This study used no GPUs or computing clusters. We ran the experiments on an Intel I9 9900k and 32GB of 3200MHz RAM running Ubuntu 20.04.3 LTS. Table A.3 shows peak RAM usage and time in seconds to completion for every subprocess of the replication.

# 4 Results

## 4.1 Quantitative Results

We started by confirming that the bias subspace does capture the difference or bias that the seed pairs are intended to represent. For this, we reproduced an experiment by Antoniak and Mimno [1] ranking the cosine similarity between

---

[5]https://github.com/theeluwin/pytorch-sgns
[6]https://github.com/ddehueck/skip-gram-negative-sampling

the first Principal Component (PC) of the bias subspace and all words in the corpus. The top and bottom ten words for each bias subspace are shown in Fig. 1a. In the shown words of the *gender pair subspace* and the *shuffled gender pair subspace* gender-related words are found, whereas none are present in the *random pair subspace*. However, only the *gender pair subspace* divides nicely between *male* and *female* terms. We extended this by calculating the cosine similarity of the top and bottom ten words from the *ordered bias subspace* for the *shuffled bias subspace*. The results in Fig. A.2 show *she* and *his* as the two highest-ranked words, which are not split along the intended bias subspace.

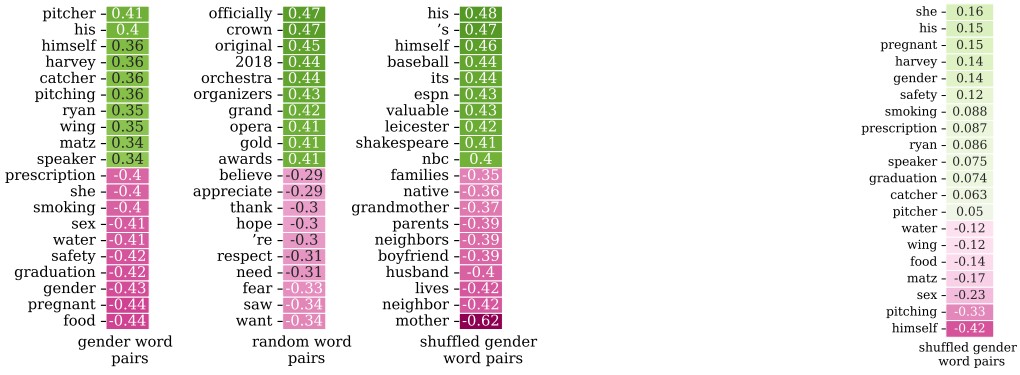

(a) Compares the top and bottom ten words of each bias subspace ranked by cosine similarity out of all words in the corpus.

(b) Top & bottom ten words of ordered subspace ranked for the shuffled subspace.

Figure 1: Replication of Fig. 4 of the original paper. Ranks words from corpus by cosine similarity against different bias subspaces (first principal component), with NYT frequency threshold 100.

Fig. 2 shows that the first PC has almost always a very high explained variance ratio for the bias subspace of ordered pairs, which drops off quickly for the subsequent PCs. Instead, the explained variance ratio per PC drops more smoothly for the shuffled pairs. Fig. A.2 shows this behavior by computing the top and bottom ten words by cosine similarity against the second PC of the gender subspace. We can observe that the bias subspace of the ordered pairs does not contain gender words anymore. In contrast, the shuffled subspace does have gender words such as *her*, thereby replicating the trend observed in Fig. 2. It is also important to note that in Fig. 2 there are exceptional cases where shuffled seed sets produce the first PC with a higher explained variance than the ordered seed sets. In general, these results replicate the trends of the original experiments.

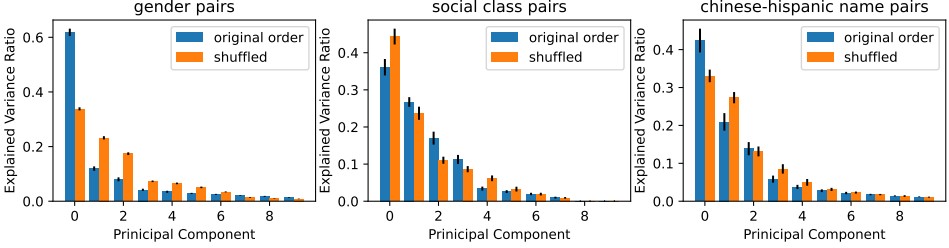

Figure 2: Replication of Fig. 3. The first ten principal components of the bias subspace for different seed pairs on the NYT corpus with a minimum frequency of 0.

Fig 3 shows that bias measurement is highly inconsistent across seed sets with the same seed category sourced from different papers. We used the cosine similarity between *female* seed sets and the word *unpleasantness* as a bias measurement. The cosine similarity varies greatly between seed sets, replicating the same trends as the original paper.

Fig. 4 explores the relationship between set similarity and the robustness of the bias subspace. The relationship between set similarity and the explained variance of the PCA-derived bias subspace vector is plotted for each dataset and frequency thresholds. The original paper shows this relationship only for the WikiText dataset, and we find a similar negative correlation between set similarity and explained variance for that dataset.

Table A.2qualitatively explores this relationship, ranking both gathered and generated sets by coherence. More semantically dissimilar seed sets score higher in coherence than more similar sets. In the gathered sets, seed sets related to names have extremely low coherence due to their semantics being very similar and the set pairs containing duplicate terms (see "names black" and "names white"). In the generated sets, we see that very different terms (such as those relating to careers and those related to lower body clothing/parts) have high coherence. In contrast, sets such as food terms score much lower. We observe a similar pattern when using the PCA algorithm as a basis for coherence. These results show the replicability of the original paper, as they are almost identical.

## 4.2 Qualitative Results

The original paper gathered 178 seed sets of eighteen highly-cited prior work on bias measurement. These seeds are both embedding-based and non-embedding-based bias detection methods, often over-lapping. The seeds are chosen in a multitude of ways. Only unigram seeds are selected, and words that do not appear in the training cor-pus are omitted. We have validated the accuracy of Table 3 in the origi-nal paper by reviewing each of the eighteen papers and determining which methods the authors used. We briefly summarize them below:

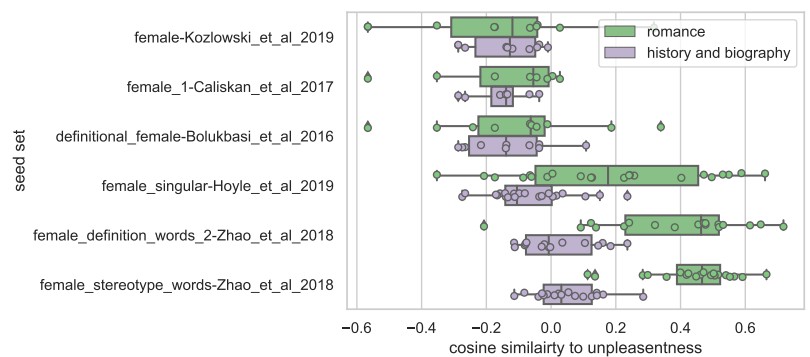

Figure 3: Reproduction of Fig. 2. Displaying the cosine similarity between the averaged vector of *unpleasantness* across all 20 bootstrapped models and different seeds sets of the category *female*.

**Borrowed from social sciences**
Select seed sets are borrowed from prior psychology and other social sciences work.

**Crowd-Sourced**  Crowd-based annotation can create custom seed sets. This method can aid in gathering contemporary associations and stereotypes. However, controlling crowd demographics often poses a problem. This can lead to stereotypes being hard-coded into the seeds.

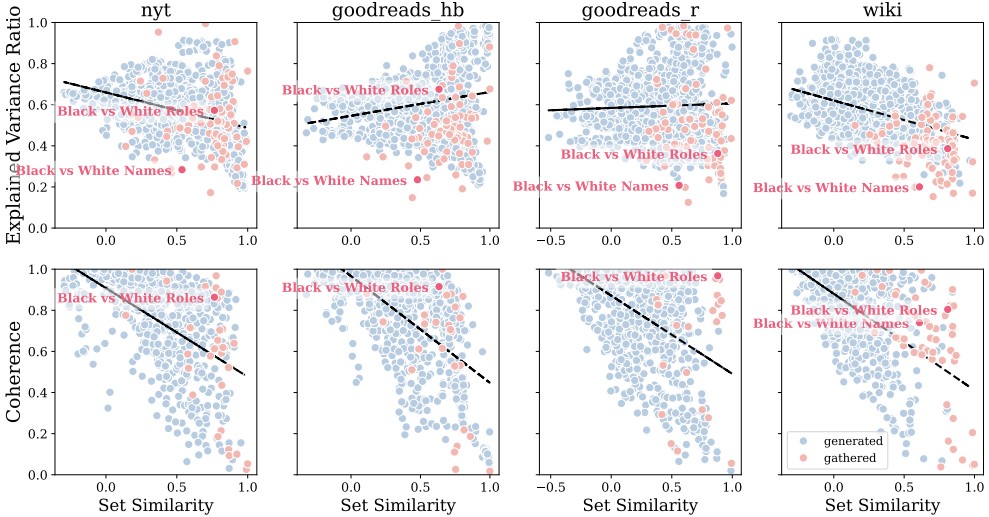

Figure 4: Replication of Fig. 5 from the original paper, displaying Explained Variance Ratio (top) and Coherence (bottom) vs Set Similarity across the four datasets. We highlight two pairs of gathered seed sets, Black vs White roles and names. For some corpora, seed terms were not found in the embeddings, causing the highlighted pair to be missing.

**Population-Derived**   Seeds can also be derived from government-collected population datasets. These datasets are usually names and occupations common to specific demographic groups. A significant problem with this method is that the data tends to be often US-centric and thus gives a distorted view of the rest of the world.

**Adapted from Lexical Resources**   Researchers can also draw seeds from existing dictionaries, lexicons and other public resources. The advantage is that these seeds have already undergone a round of validation.

**Corpus-Derived**   This quantitative method is used to extract seeds terms from a corpus. It has the advantage of ensuring high-frequency words are selected but suffers from similar risks as crowd-sourced seeds.

**Curated**   Seed hand-selection by authors often yields high precision seeds but is slow and relies on unbiased authors.

**Re-used**   The last method relies on prior bias measurement research for seed terms. The advantage is that the seeds have already been used, but researchers should not use them without validation.

### 4.3   Results beyond original paper

**Set Similarity and Bias Subspace in Additional Datasets**   We extended the original paper's set similarity versus bias subspace explained variance analysis to cover all datasets (beyond WikiText) in Fig. 4. The negative trend is still present with the NYT corpus, but not in the Goodreads corpora, where the trend is almost absent or slightly positive. In addition, the positions of the highlighted seed set pairs are variable across corpora. We also extended this work to examine the relationship between seed pair coherence versus set similarity, where the inverse relationship is present in all datasets. Notice that the requirement that coherence is calculated only for models that contain all seed terms (as described in Section 3.5) makes specific pairs of seed sets be ignored, as seen from the lack of the two highlighted set pairs for select datasets.

**Testing Minimum Frequency Filter**   Due to inconsistencies both in the paper and in communication with the author in the reported minimum frequency filter for the skip-gram models, we experimented with minimum frequencies $\mu \in \{0, 10, 100\}$. These enabled us to see results across the whole vocabulary in the case of $\mu = 0$ and reduce noise from rare words in the case of $\mu = 10$. We also used $\mu = 100$ to generate Fig. 1 as the original paper.

**Seed Toolkit and Pairing Seed Set Data.**   Other than extending the experiments of the original paper, we have two additional contributions. For the sake of reproducibility, we make our code publicly available and design our repository as an open Python package that can be used to obtain bias subspace vectors and assess seed set robustness. This toolkit can help future researchers who aim to evaluate their seeds carefully. Our second contribution is an augmentation of the seed dataset provided by Antoniak and Mimno [1]. We provided additional annotations regarding pairing, i.e. we identify which seeds to pair together along standard bias dimensions in a queriable `.csv` format.

## 5   Discussion

Overall, our results replicate the data reported in the original paper. This replication lends strong support to the general claim of the original paper that seed sets incorporate strong inductive biases that affect their performance as grounding for bias metrics and that researchers should be more cognizant of these limitations.

Instability in bias subspaces can be introduced by selecting seeds in seed sets, as stated in claim 2b. Our results in Fig. 3 support this as they reproduce the original work. The same bias measurement varies across seed sets selected by different authors who assigned it to the same category. In addition, the dependence of the bias subspace on seed set selection is further supported by Fig. 4. The two highlighted seed sets (black vs white roles/names) are generally distinct in position for each corpus, despite theoretically attempting to define similar bias dimensions.

Another source of instability claimed by Antoniak and Mimno [1] is the ordering and pairing of seed sets. In Fig. 2 we show that the explained variance ratio for the *ordered bias subspaces* can behave very differently from the *shuffled bias subspaces*, supporting claim 2a. Our work in Fig. 1a also supports this claim. While the ordered subspace successfully splits the top words along the intended subspace of *male* and *female*, the first PC of the shuffled bias subspace has words such as *mother* and *boyfriend* both ranked on the same end. This shows that while the subspace still picks up on

gender words, it does not represent the intended subspace. Supporting claim 2a that bias subspaces can become less meaningful with a shuffled seed pairing. We could further confirm this behavior by calculating the cosine similarity of the top words of the *ordered subspace* for the *shuffled subspace* in Fig. 1b. These results show that *she* and *his* are ranked next to each other at the top and not split along the intended bias subspace. These experiments lend strong support to claim 2a that the order of seed pairs can substantially influence the meaningfulness of the bias subspace and, consequently, the bias metrics.

Finally, bias subspaces suffer instability due to semantically overlapping seeds being less distinguishable in the bias subspace, as stated in claim 2c. Our results in Table A.2 and Fig. 4 demonstrate that bias subspace vectors are less robust when the seed sets are semantically similar or overlapping. This relationship lends strong credence to claim 2c. However, our results did show that this inverse relationship is not conserved across a minority of corpora (e.g., the Goodreads datasets) for the explained variance metric. More broadly, however, this still shows that the reliability of seed selection is quite variable. While similar seed sets may generate robust bias subspaces for more semantically equivalent seed pairs for some corpora, that is not guaranteed. Therefore, while this inverse relationship may be minimized for specific corpora, extensive corpus-specific seed set investigations are still required.

**What was easy.** The original paper clearly described the algorithms used to obtain bias metrics. Additionally, it carefully cited the papers that first proposed them, which specified further details. This aided our understanding of the underlying concepts and accelerated the implementation of the frameworks. Model training and embedding generation was also facilitated by the pre-existing *gensim* framework. This permitted greater focus on reproducing the details of the experiments than choosing between alternative implementations of skip-gram word2vec. In addition, responsive authors permitted quick clarifications through email communication when important details were not clear.

**What was difficult.** The original paper did not make code publicly available and largely lacked documentation. Only the gathered seeds were provided via GitHub (2). This made it necessary to reproduce all the code from scratch.

In select instances, the paper crucially omitted important information, making us reliant on communication with the authors. This was most pronounced when aggregating embeddings or other metrics across the bootstrapped model sampling, where vocabulary sizes were different. This meant that not all models had good embeddings for all seed terms. We had to consider several different approaches before settling on the averaging criteria described in Section 3.5.

Finally, preprocessing the data was more difficult than initially imagined. The tokenization pipeline in the original paper was vaguely specified, and differences in our implementation caused the slight discrepancies in Table A.3. The POS tagging with spaCy was imperfect, resulting in the incorrect tagging of several proper nouns as common nouns, making it hard to control for POS in random seed generation.

**Communication with original authors.** While the authors did not disclose any code, we maintained a lengthy email correspondence with them. One author, Maria Antoniak, was contacted to clarify hyperparameters of the word2vec model, the methodology for generating random seeds across bootstrapped models, and which bias metrics (PCA or WEAT) were used for different results. She also described her dataset processing pipeline, as there were many alternate ways to process the corpora before training.

## 6 Conclusion

Overall, our results replicate the ones reported in the original paper. This lends strong support to the general claim of the original paper that seed sets incorporate significant inductive biases that affect their performance as grounding for bias metrics and that researchers should be more cognizant of these limitations. Aside from confirming the danger of blindly using seed sets, we also provide additional contributions. First of all, all code used to replicate the original paper is publicly available. This code can obtain bias subspace vectors and assess seed set robustness. Secondly, we extended the original paper's set similarity versus bias subspace explained variance analysis to cover all datasets. Furthermore, we implement multiple numbers of minimum frequencies that further enable results across the entire vocabulary. Lastly, we provide an additional annotation pairing of the original seed dataset.

We have highlighted a need for carefully justifying the use of particular sets through empirical means, but a theoretically sound and systematic method for doing so is still in its infancy. Further work may explore what criteria seed sets should satisfy to demonstrate robustness. In addition, future researchers may want to extend this work to bigram seed terms and embeddings to explore the limitations of more expressive seeds and bias dimensions.

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

 # A   Appendix

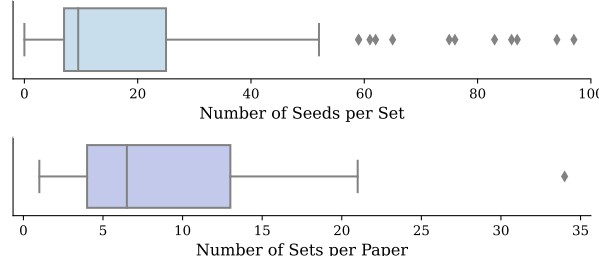

Figure A.1:  Replication of Fig. 1 from the original paper, illustrating basic statistics of the gathered seeds.

Table A.1: Comparing corpora summary statistics after preprocessing (original paper statistics obtained from Table 2).

| Dataset | Total Documents | | Total Words | | Vocabulary Size | | Mean Document Length | |
|---|---|---|---|---|---|---|---|---|
| | original | ours | original | ours | original | ours | original | ours |
| **NYT** | 8,888 | 8,888 | 7,244,457 | 7,217,851 | 162,998 | 109,713 | 815 | 812 |
| **WikiText** | 28,472 | 28,472 | 99,197,146 | 87,077,718 | 546,828 | 228,318 | 3,484 | 3,058 |
| **Goodreads (Romance)** | 197,000 | 194,500 | 24,856,924 | 24,695,141 | 214,572 | 249,114 | 126 | 127 |
| **Goodreads (History/Biog)** | 136,000 | 135,000 | 14,324,947 | 14,168,742 | 163,171 | 193,012 | 105 | 105 |

Table A.2:  Replication of Table 4 from the original paper.  Seeds that are more semantically similar have lower coherence scores. We use the WEAT metric (the difference between the mean vectors of the seed sets) to generate the subspace and the NYT dataset embeddings for this data. We average coherence scores across the $n$ models (out of 20) that contain the paired seed sets and round to 3 decimal places. Unfortunately, while we tried to limit generated sets to only common nouns, proper nouns and, more rarely, verbs appeared in the sets due to issues with the spaCy POS tagger.

| Coherence | Generated Set A | Generated Set B |
|---|---|---|
| 1.000 | know, believe, think, guess, mean | governor, mayor, legislature, senator, democrat |
| 1.000 | foot-8, foot-7, foot-3, foot-5, to-4 | rousteing, atkins, cornejo, ehrenreich, yorke |
| 0.999 | associate, assistant, economist, engineer, accountant | heels, shoes, pants, legs, fingers |
| ... | ... | ... |
| 0.062 | hertl, agnieszka, goran, brouwer, koivu | bases, wings, outs, scoreless, rockies |
| 0.059 | molina, glasser, pitney, darren, mackenzie | carver, mina, boyce, curator, deputy |
| 0.053 | lime, juice, lemon, potato, garlic | combo, bodysuit, raisin, koji, mango |

| Coherence | Gathered Set A | Gathered Set B |
|---|---|---|
| 0.999 | CAREER: executive, management, professio... | FAMILY: home, parents, children, famil... |
| 0.968 | MALE: brother, father, uncle, grandfat... | FEMALE: sister, mother, aunt, grandmot... |
| 0.942 | TERRORISM: terror, terrorism, violence,... | OCCUPATIONS: banker, carpenter, doctor,... |
| ... | ... | ... |
| 0.093 | MALE NAMES: john, paul, mike, kevin, ... | FEMALE NAMES: amy, joan, lisa, sarah,... |
| 0.053 | NAMES BLACK: harris, robinson, howard, ... | NAMES WHITE: harris, nelson, robinson, ... |
| 0.026 | NAMES ASIAN: cho, wong, tang, huang, ... | NAMES CHINESE: chung, liu, wong, huang... |

Table A.3: Computing power needed for each action in the replication process.

| Action | Time (s) | RAM (MB) |
|---|---|---|
| Downloading the data | 293 | 427 |
| Preprocessing the data | 3054 | 19018 |
| Training all models | 7806 | 21054 |
| Table A.1 | 4274 | 661 |
| Fig. 1 | 22 | 4363 |
| Fig. 2 | 19 | 4370 |
| Fig. 3 | 4 | 1510 |
| Fig. 4 | 500 | 1610 |

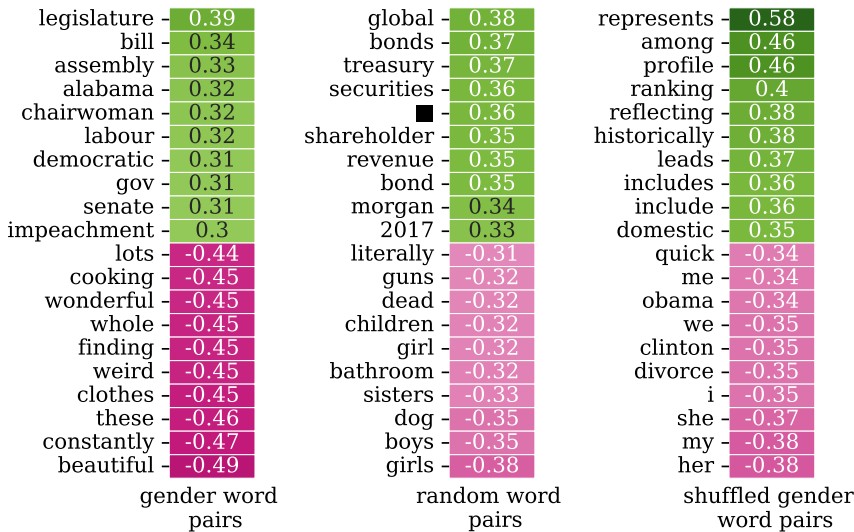

Figure A.2: Extension on Fig. 4 from the original paper. Ranks words from the NYT corpus by cosine similarity against different bias subspaces (2nd principal component), with NYT frequency threshold 100.

Table A.4: Seeds used in various Figures

| Figures | Seed ID | Seeds |
|---|---|---|
| Fig. 3 | female-Kozlowski_et_al_2019 | ['woman', 'women', 'she', 'her', 'her', 'hers', 'girl', 'girls', 'female', 'feminine'] |
| | female_1-Caliskan_et_al_2017 | ['sister', 'female', 'woman', 'girl', 'daughter', 'she', 'hers', 'her'] |
| | definitional_female-Bolukbasi_et_al_2016 | ['woman', 'girl', 'she', 'mother', 'daughter', 'gal', 'female', 'her', 'herself', 'mary'] |
| | female_singular-Hoyle_et_al_2019 | ['woman', 'girl', 'mother', 'daughter', 'sister', 'wife', 'aunt', 'niece', 'empress', 'queen', 'princess', 'duchess', 'lady', 'dame', 'waitress', 'actress', 'goddess', 'policewoman', 'postwoman', 'heroine', 'witch', 'stewardess', 'she'] |
| | female_definition_words_2-Zhao_et_al_2018 | ['lady', 'saleswoman', 'noblewoman', 'hostess', 'coquette', 'nun', 'heroine', 'actress', 'chairwoman', 'businesswoman', 'spokeswoman', 'waitress', 'councilwoman', 'stateswoman', 'policewoman', 'country-women', 'horsewoman', 'headmistress', 'governess', 'widow', 'witch', 'fiancee'] |
| | female_stereotype_words-Zhao_et_al_2018 | ['baker', 'counselor', 'nanny', 'librarians', 'socialite', 'assistant', 'tailor', 'dancer', 'hairdresser', 'cashier', 'secretary', 'clerk', 'stenographer', 'optometrist', 'housekeeper', 'bookkeeper', 'homemaker', 'nurse', 'stylist', 'receptionist'] |

| | | |
|---|---|---|
| Fig. 2 | definitional_female-Bolukbasi_et_al_2016 | ['woman', 'girl', 'she', 'mother', 'daughter', 'gal', 'female', 'her', 'herself', 'mary'] |
| | definitional_male-Bolukbasi_et_al_2016 | ['man', 'boy', 'he', 'father', 'son', 'guy', 'male', 'his', 'himself', 'john'] |
| | definitional_female-Bolukbasi_et_al_2016 shuffled | ["herself", "woman", "daughter", "mary", "her", "girl", "mother", "she", "female", "gal"] |
| | definitional_male-Bolukbasi_et_al_2016 shuffled | [ "man", "his", "he", "son", "guy", "himself", "father", "boy", "male", "john"] |
| | upperclass-Kozlowski_et_al_2019 | ['rich', 'richer', 'richest', 'affluence', 'affluent', 'expensive', 'luxury', 'opulent'] |
| | lowerclass-Kozlowski_et_al_2019 | ['poor', 'poorer', 'poorest', 'poverty', 'impoverished', 'inexpensive', 'cheap', 'needy'] |
| | upperclass-Kozlowski_et_al_2019 shuffled | ["richer", "opulent", "luxury", "affluent", "rich", "affluence", "richest", "expensive" ] |
| | lowerclass-Kozlowski_et_al_2019 shuffled | [ "poorer", "impoverished", "poorest", "cheap", "needy", "poverty", "inexpensive", "poor"] |
| | names_chinese-Garg_et_al_2018 | ['chung', 'liu', 'wong', 'huang', 'ng', 'hu', 'chu', 'chen', 'lin', 'liang', 'wang', 'wu', 'yang', 'tang', 'chang', 'hong', 'li'] |
| | names_hispanic-Garg_et_al_2018 | ['ruiz', 'alvarez', 'vargas', 'castillo', 'gomez', 'soto', 'gonzalez', 'sanchez', 'rivera', 'mendoza', 'martinez', 'torres', 'rodriguez', 'perez', 'lopez', 'medina', 'diaz', 'garcia', 'castro', 'cruz'] |
| | names_chinese-Garg_et_al_2018 shuffled | ["tang", "chang", "chu", "yang", "wu","hong", "huang", "wong", "hu", "liu", "lin", "chen", "liang", "chung", "li", "ng", "wang"] |
| | names_hispanic-Garg_et_al_2018 huffled | ["ruiz", "rodriguez", "diaz", "perez", "lopez", "vargas", "alvarez", "garcia","cruz", "torres", "gonzalez", "soto", "martinez", "medina", "rivera", "castillo", "castro", "mendoza", "sanchez", "gomez"] |
| Fig. 1 Fig.A.2 | definitional_female-Bolukbasi_et_al_2016 | ['woman', 'girl', 'she', 'mother', 'daughter', 'gal', 'female', 'her', 'herself', 'mary'] |
| | definitional_male-Bolukbasi_et_al_2016 | ['man', 'boy', 'he', 'father', 'son', 'guy', 'male', 'his', 'himself', 'john'] |
| | definitional_female-Bolukbasi_et_al_2016 shuffled | ["female", "she", "woman", "gal", "her", "daughter", "girl", "herself", "mother", "mary"] |
| | definitional_male-Bolukbasi_et_al_2016 shuffled | ["john","man", "son","father", "male","himself", "guy","he", "his","boy"] |
| | random seeds 1 | ['essential', 'want', 'suspension', 'talked', 'competitive', 'information', 'hero', 'bat', 'seconds', 'black'] |
| | random seeds 2 | ['derby', 'passed', 'achieve',, 'discussed', 'providing', 'resulted', 'inmates', 'wearing', 'bid', 'rose'] |

