# OpenReview forum: "Badder Seeds: Reproducing the Evaluation of Lexical Methodsfor Bias Measurement"
_ML_Reproducibility_Challenge/2021/Fall — RC2021_

### Official Review · Reviewer_oSyS · 2022-03-01
**Good effort reimplementing**

**Rating:** 7
**Confidence:** 3

**Review:**

Reproducibility Summary:
The paper has this which states the scope, results, methodology, what is easy and hard and communication with authors.

Scope of reproducibility:
The study aims to evaluate the reproducibility of the quantitative and qualitative results presented in the paper and the conclusions drawn thereof.

Code:
Since no code was made public, the authors had to reimplement everything in the study. The code is made available for review: https://anonymous.4open.science/r/mlrc-2021-A0C2

It is puzzling why the authors did not release the code.

Communication with original authors:
The original authors were very helpful and engaged in length email exchanges.

Ablation Study:
Both this work and the original paper do not provide ablation study.

Discussion on results:
The authors reproduce most of the results supporting the original authors’ general claim: seed sets often suffer from biases that affect their performance as a baseline for bias metrics. However, they are slightly different on select occasions, but not in ways that undermine the paper’s general intent to show the fragility of seed sets.

Recommendations for reproducibility:
The paper makes recommendation on (1) a need for carefully justifying the use of particular sets through empirical means (2) further work exploring what criteria seed sets should satisfy to demonstrate robustness (3) extend this work to bigram seed terms and embeddings to explore the limitations of more expressive seeds and bias dimensions.

The paper also goes beyond reproducing results: (1) extended the original paper’s set similarity versus bias subspace explained variance analysis to cover all datasets  (2) testing Minimum Frequency Filter (3) providing seed toolkit and pairing seed set data.

---

### Official Review · Reviewer_SFZ3 · 2022-03-17
**Badder Seeds: Reproducing the Evaluation of Lexical Methodsfor Bias Measurement**

**Rating:** 6
**Confidence:** 4

**Review:**

The authors propose a replication of Antoniak and Mimno's experiment. This work was initiated by the fact that the original paper is not publicly available, which led the authors to completely reimplement the code. The entire source code is freely available to the scientific community via the GitHub platform. In order to respect the reference article, they have studied 5 datasets initially present in the original study (New York Times, WikiText-103, Goodreads, GoogleNews, and Seed Set Dataset). Two metrics were used, namely (WEAT and PCA), to respect and compare the source work.
The basic approach is interesting, but I think the introduction/background and the method section could both be greatly improved.

* The original study had more data sets, so the authors chose the 5 data sets. No explanation is given for this.
* In Figure 1, the authors should explain why gender-related words are present in the gender pair subspace and in the mixed-gender pair subspace, while none are present in the random pair subspace.
* Why does the cosine dissimilarity value of female_singular-Hoyle_and_al_2019 show a large variability (see Figure 3)? Can you explain?

Minor corrections:

- Line 37: an empirical manner. [11, 6] Crucially -> an empirical manner [11, 6]. Crucially
- Line 76: seed sets gathered from prior works 2 -> no space between works and 2
- Line 88: GoogleNews model 4 -> no space between model and 4
- Page 3 (footage): 2 https://github.com/maria-antoniak/bad-seeds -> no space between 2 and https
- Line 125: GitHub if that is preferred. 5,6. -> GitHub if that is preferred5,6.
- Line 177: Table A.2qualitatively -> Table A.2 qualitatively

---

### Meta-Review · Area_Chair_gMf2 · 2022-04-09

**Recommendation:** Accept
**Confidence:** 4

**Metareview:**

Reviewers praised that the authors reproduced code that wasn't available. They also praise the experiments.

---

### Decision · Program_Chairs · 2022-04-09

**Decision:**

Accept

**Comment:**

Following the recommendation of reviewers and meta-reviewer, the paper is accepted for ML Reproducibility Challenge 2021, and will be published in the upcoming special edition of ReScience Journal.